# FGFR2 overexpression and compromised survival in diffuse-type gastric cancer in a large central European cohort

**Thorben Schrumpf, Hans-Michael Behrens, Jochen Haag, Sandra Krüger, Christoph Röcken** *

Dept. of Pathology, Christian-Albrechts-University, University Hospital Schleswig-Holstein, Kiel, Germany

* christoph.roecken@uksh.de

**Data Availability Statement:** The study's minimal underlying data are provided in S1 Table (Supplementary Tables_R1_xlsx).

## Abstract

The significance of fibroblast growth factor receptor 2 (FGFR2) in gastric cancer (GC) has been studied predominantly in Asian patient cohorts. Data on White patients are scarce. Here, we aimed to independently validate the expression and putative tumor biological significance of FGFR2 in a large non-Asian GC cohort. Immunohistochemistry (IHC) was performed on large-area tissue sections from 493 patients with GC and evaluated using the HScore. GCs with moderate and strong FGFR2 expression were studied for *Fgfr2* amplification using chromogenic in situ hybridization (CISH). Median overall survival was determined using the Kaplan–Meier method. The majority [240 (99.1%)] of FGFR2-positive GCs showed a variable combination of staining intensities with marked intratumoral heterogeneity, including weak [198 (40.2%) cases], moderate [145 (29.4%)], and strong [108 (21.9%)] staining in diverse combinations. 250 (50.9%) GCs expressed no FGFR2. *Fgfr2* gene amplification was found in 40% of selected cases with high protein expression and was also heterogeneous at the cell level. FGFR2 protein expression did not correlate with patient survival in the entire cohort However, using different cutoff values, a negative correlation between FGFR2-expression and patient outcome was found for diffuse-type GC. FGFR2 expression was associated with a lower tumor grade and intestinal phenotype (p≤0.0001). FGFR2–positive diffuse-type GCs classify a small subset of patients with a poor tumor specific survival (5.29±1.3 vs. 14.67±1.9 months; p = 0.004).

## Introduction

Despite the declining incidence of gastric cancer (GC) in recent decades, it remains the fifth most common malignancy and third leading cause of cancer-related deaths worldwide [1]. Its incidence remains high, especially in Asia [2]. At the time of diagnosis, approximately two-thirds of the patients present with an advanced disease stage [3]. However, treatment options are limited. Curative surgery is no longer an option in most patients. Palliative chemotherapy and supportive therapy remain the only available treatment [4]. Amplification of different tyrosine kinase receptors (TKRs) has been described in GC [5]. However, until today

**Funding:** The author(s) received no specific funding for this work.

**Competing interests:** The authors have declared that no competing interests exist.

treatment with epidermal growth factor receptor (ERBB2 or Her2/neu) inhibitors remains the only approved first-line targeted tyrosine kinase inhibitor (TKI) therapy [6], and *HER2* amplification occurs in only about 8% of GC patients [7]. Treatment with other targeted therapies, such as ramucirumab, pembrolizumab, or nivolumab (anti-PD-1) remains a second- and third-line therapeutic option for patients with GC. Despite this progress, the overall survival prognosis of patients with advanced GC remains poor [8].

Alterations in the fibroblast growth factor receptor (FGFR) pathway have been investigated as therapeutic targets for diverse tumor types [9]. The potential prognostic significance of GC was first described in 1994 [10]. Amplification of *Fgfr2* in GC has been shown to be an independent prognostic factor for patient survival [11]. The prevalence of *Fgfr2* gene amplification has been reported with 2–9% in GC patients [5, 11–16]. FGFR2 protein expression in GC has been investigated in several occasions [17–27]. Overexpression of FGFR2 was reported in as many as 60% of the patients [25, 27]. There have been various reports on the significance of FGFR2 overexpression. Initially, there were indications that high expression was associated with better patient outcome [18]. Most studies have demonstrated FGFR2 overexpression as a prognostic marker for poor overall survival (OS) or tumor-specific survival (TSS) [20, 22, 23, 27]. Another study demonstrated prognostic significance only in patients with diffuse-type GC [26]. A meta-analysis published in 2019 concluded that high FGFR2 protein expression in GC was associated with worse outcomes, greater depth of invasion, higher rates of lymph node metastasis, and more advanced disease stage [28]. However, the current data on FGFR2 protein expression in GC are predominantly from Asian study populations, and the meta-analysis identified a gap for White patients [28]. In order to fill this gap in information, we studied FGFR2-status and its correlation with diverse clinicopathological patient characteristics in a large cohort of White GC patients.

## Material and methods

### Ethics approval and consent to participate

The study was carried out in accordance with the ethical standards of the responsible committee on human experimentation (institutional and national) and the Helsinki Declaration of 1964 and later versions. This study was approved by the local ethics committee of the University Hospital in Kiel, Germany (reference number D 453/10).

### Study population

From 1997 to 2009, we identified all White patients who had undergone either total or partial gastrectomy for adenocarcinomas of the stomach or esophagogastric junction at the University Hospital Kiel (GC cohort). The following patient characteristics were documented: type of surgery, age at diagnosis, sex, tumor localization and size, tumor type, tumor grade, depth of invasion, number of lymph nodes resected, and number of lymph nodes with metastases. The date of patient death was obtained from the Epidemiological Cancer Registry of the State of Schleswig-Holstein, Germany. The follow-up data of patients who were still alive were retrieved from hospital records and interviews with general practitioners.

### Study inclusion and exclusion criteria

Patients were included and excluded according to the following criteria. Patients were included when histology confirmed adenocarcinoma of the stomach or esophagogastric junction, and data on death or survival were available. Patients were excluded if histology identified a tumor type other than adenocarcinoma, histopathological data were incomplete, patients

had previously undergone a partial gastrectomy (Billroth II), and had locally recurrent GC or data on patient death or survival could not be obtained. Patients who had received neoadjuvant or perioperative chemotherapy were excluded from the study. After inclusion in the study every related patient data was pseudonymized.

## Histology and TNM classification

Tissue specimens had been fixed in 10% neutral buffered formalin and embedded in paraffin. Formalin fixation was standardized during the study period. Deparaffinized sections were stained with hematoxylin and eosin. Tumors were classified according to the Laurén classification [29]. The pTNM stage of all study patients was determined according to the eighth edition of the UICC guidelines [30] and was based solely on surgical pathological examination, including the classification of distant metastases (pM-category). Patients previously enrolled in the study were re-categorized accordingly.

## Immunohistochemistry

The expression level of FGFR2 was assessed by immunohistochemistry (IHC). Monoclonal anti-FGFR2 antibody with a dilution of 1:20 (ab10648, Abcam®) and the Autostainer Bond Max System (Leica Microsystems GmbH, Wetzlar, Germany) were used. Antigen retrieval was performed using the ER1 citrate-buffer antigen retrieval solution for 20 min at pH 6.0 (Leica-Menarini). The Bond Polymer Refine Detection Kit (Leica Biosystems) was used for antigen detection.

Two independent observers assessed FGFR2 immunostaining (FGFR2-IHC). Both observers were blinded with regard to clinicopathological patient characteristics. The entire cohort was screened and membranous, cytoplasmic and nuclear staining was documented. However, intensity of immunostaining between the three different cellular compartments did not vary distinctively enough, to allow a separate evaluation and therefore the expression level was categorized "globally" into four different grades (FGFR2-IHC 0, $1^+$, $2^+$, and $3^+$) (S1 Table). FGFR2-IHC 0 was characterized by a complete lack of tumor cell immunostaining. FGFR2-IHC $1^+$ was characterized by faint immunostaining, whereas FGFR2-IHC $2^+$ and FGFR2-IHC $3^+$ were characterized by strong immunostaining. To ensure consistent evaluation for all cases, representative samples of each expression level were obtained (Fig 1) and subsequently used as a reference standard for in-depth evaluation of the entire cohort. FGFR2 expression was evaluated according to the HScore as described previously [31]. In brief, the percentage of positive tumor cells showing the defined staining intensities (0, 1+, 2+, 3+) was increased with respect to all tumor cells visible on each tissue specimen, and it always added up to a total of 100% tumor cells. The HScore was then calculated according to the following formula: HScore = [0×percentage of immunonegative tumor cells]+[1×percentage of weakly stained tumor cells]+[2×percentage of moderately stained tumor cells]+[3×percentage of strongly stained tumor cells], resulting in a possible HScore between 0 and 300. Tumor cells without detectable staining were scored 0. The maximum possible HScore was 300 if all cells of a given tumor sample showed strong staining: [0×0%]+[1×0%]+[2×0%]+[3×100%] = 300.

We documented the heterogeneity of FGFR2 distribution inside the tumor, the localization of the stained cells inside the tumor, the staining of non-neoplastic tissue, the localization of FGFR 2 staining inside the tumor cells (membranous, cytoplasmic, nuclear; S1 Table), and the presence of FGFR2 stained cells undergoing apoptosis.

## Chromogenic in-situ hybridization (CISH)

Analysis of *Fgfr2* amplification was analyzed by CISH using the ZytoDot® 2C (SPEC *Fgfr2*/CEN 10 Probe) and the ZytoDot 2C CISH Implementation Kit (ZytoVision GmbH,

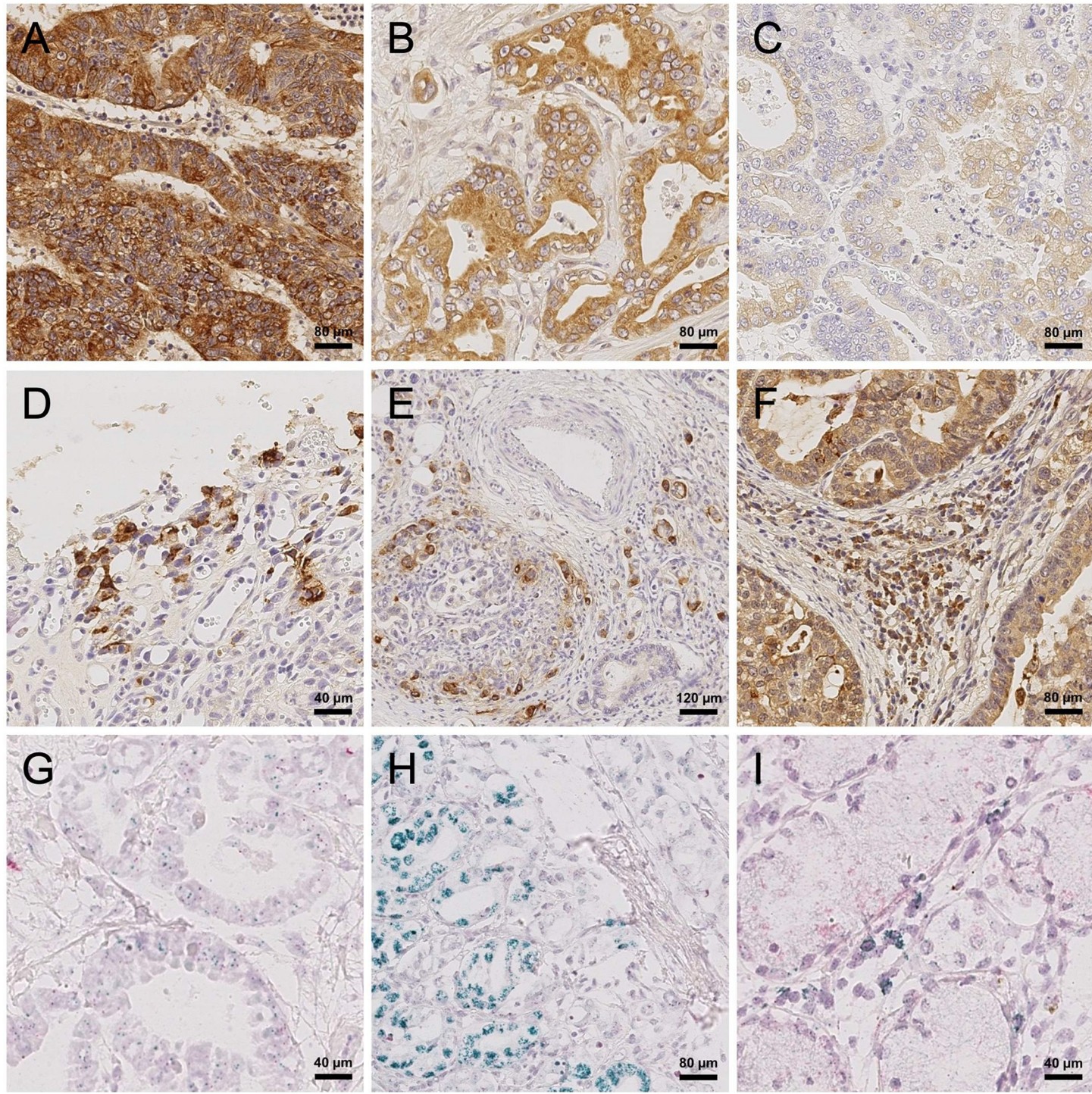

**Fig 1. FGFR 2—protein expression and gene amplification in gastric cancer.** FGFR2 protein expression was examined by immunohistochemical staining of the FGFR2 receptor. Representative cases with intensity of FGFR-IHC 3+ (A), FGFR-IHC 2+ (B), and FGFR-IHC 1+ (C) were selected and used as reference throughout the study. During immunohistochemical examination of FGFR2, the vast majority of cases with increased FGFR2 protein expression showed a heterogeneous distribution of protein expression (D-E). Strongly FGFR2-positive stromal cells were apparent in 92 (18.5%) cases during the study (F). Sixty cases with strong FGFR2 protein expression were examined for *Fgfr2* amplification by chromogenic in situ hybridization. The signals of the *Fgfr2* genes showed a green signal, centromere 10 signals a red signal (G). Clusters of *Fgfr2* amplifications in tumor cells were found in 18 cases (H). Clusters of *Fgfr2* amplifications appeared in close proximity to nonamplified tumor cells (H). Nontumor cells with increased *Fgfr2* signals were observed in 14 cases (I). Original magnification: 100× (E), 200× (A, B, C, F, H), 400× (D, G, I).

Bremerhaven, Germany). *Fgfr2*-CISH was evaluated by screening the entire tissue section to find *Fgfr2* amplified tumor cells. *Fgfr2* and centromere 10 signals were counted in at least 20 representative adjacent tumor cell nuclei within the invasive regions. The *Fgfr2*/centromere ratio of 10 was then calculated. Probes with a ratio greater than 2.2 were classified as *Fgfr2* amplified. Probes with a ratio lower 1.8 were classified as non-amplified. Forty nuclei were counted if the *Fgfr2*/centromere 10 ratio ranged from 1.8 to 2.2. If the ratio was still in the range of 1.8 to 2.2, a cutoff of >2.0 was used to classify probes as *Fgfr2* amplified. CISH clusters were also observed. Tumors with an average *Fgfr2* count of greater than 4 signals per nucleus were classified as *Fgfr2* polysomic.

### Assessment of phenotype, genotype and infectious status

Helicobacter pylori, Epstein–Barr virus, microsatellite, MET, HER2, PD-1, PD-L1, and VISTA status were assessed as described in detail previously [32–36].

### Statistics

Statistical analyses were performed using SPSS version 20.0 (IBM Corporation, Armonk, NY, USA). For continuous variables, cases were divided into two groups by splitting the median value. For ordinal variables, the cases were divided into categories or combinations of different categories. The median overall survival was determined using the Kaplan–Meier method, and the log-rank test was used to determine statistical significance. For comparison purposes, the median survival time, its standard deviation, and 95% confidence interval (CI) were calculated. The Mann–Whitney U test was used to investigate the association between *the Fgfr2*-CISH status and HScore. The statistical significance of the correlation between clinicopathological parameters and biomarker expression was tested using Pearson´s $Chi^2$ test. For parameters of the ordinal scale (pT category, pN category, UICC tumor stage), we applied Kendall's tau test instead. Statistical significance was set at $p \leq 0.05$. To account for the effects of multiple testing, we applied the explorative Simes (Benjamini–Hochberg) procedure. P-values are given unadjusted but are marked where they hold significance under the explorative Simes procedure. Survival times are given in months throughout the study.

### Results

A total of 493 patients fulfilled all study criteria. Clinicopathological characteristics of the patients are summarized in Table 1. Among them, 22 were EBV-positive and 36 were MSI. According to Laurén et al., 255 GCs showed intestinal, 154 diffuse, 31 mixed, and 53 unclassifiable phenotypes.

Data of overall (OS) and tumor specific survival (TSS) were available for 473 (95.9%) and 443 (89.9%) of the 493 cases, respectively. The mean follow-up period was 12.8 months (range: 0–142.7 months). The median OS was 14.9 months, and the median TSS was 16.6 months.

### Expression of FGFR2 in gastric cancer

FGFR2 expression was studied by immunohistochemistry using large-area tissue sections. Weak immunostaining (FGFR2-1[+]) was observed in 198 (40.2%) cases, moderate (FGFR2-2[+]) in 145 (29.4%), and strong (FGFR2-3[+]) in 108 (21.9%). In 50 cases with strong immunostaining (FGFR2-3[+]), only a few cells ($\leq$1%) of the tumor were stained (Fig 1). No immunostaining (FGFR2-0) of a portion of the tumor was found in 491 (99.6%) GCs. A complete lack of FGFR2 in the entire tumor area was observed in 251 (50.9%) GCs (S1 Table).

**Table 1. Correlation of clinicopathological patient characteristics with FGFR2 protein expression.**

| Cohort | | | All | | | Intestinal | | | Diffuse | | | Mixed | | | Unclassified | | |
|---|---|---|---|---|---|---|---|---|---|---|---|---|---|---|---|---|
| | | | | | | | | Tumor type according to Lauren | | | | | | | | | |
| FGFR status | | Neg. | Pos. | p | Neg. | Pos. | p | Neg. | Pos. | p | Neg. | Pos. | p | Neg. | Pos. | p |
| **Sex** | F | 104 | 80 | 0.063 | 24 | 46 | 0.546 | 67 | 21 | 0.851 | 4 | 3 | 0.685 | 9 | 10 | 1.00 |
| | M | 147 | 162 | | 71 | 114 | | 49 | 17 | | 11 | 13 | | 16 | 18 | |
| **Age** | <68 J | 129 | 112 | 0.319 | 36 | 64 | 0.691 | 74 | 23 | 0.844 | 5 | 8 | 0.473 | 14 | 17 | 0.785 |
| | >68 J | 120 | 126 | | 59 | 93 | | 40 | 14 | | 10 | 8 | | 11 | 11 | |
| **Localization** | Prox. | 66 | 82 | 0.075 | 39 | 60 | 0.589 | 16 | 4 | 0.783 | 3 | 7 | 0.252 | 8 | 11 | 0.775 |
| | Dist. | 178 | 154 | | 51 | 94 | | 98 | 34 | | 15 | 16 | | 17 | 17 | |
| **Laurén phenotype** | Int. | 95 | 160 | <.001* | | | | | | | | | | | | |
| | Diff. | 116 | 38 | | | | | | | | | | | | | |
| | Mixed. | 15 | 16 | | | | | | | | | | | | | |
| | Uncl. | 25 | 28 | | | | | | | | | | | | | |
| **pT category** | 1 | 33 | 28 | 0.358 | 21 | 25 | 0.586 | 12 | 1 | 0.461 | 0 | 1 | 0.872 | 0 | 2 | 0.405 |
| | 2 | 24 | 33 | | 11 | 25 | | 8 | 4 | | 7 | 5 | | 5 | 3 | |
| | 3 | 97 | 98 | | 37 | 64 | | 44 | 14 | | 8 | 10 | | 9 | 15 | |
| | 4 | 97 | 82 | | 26 | 45 | | 52 | 19 | | 15 | 16 | | 11 | 8 | |
| **pN category** | 0 | 70 | 69 | 0.787 | 34 | 55 | 0.991 | 27 | 6 | 0.447 | 1 | 1 | 0.520 | 8 | 7 | 0.329 |
| | 1 | 40 | 32 | | 17 | 24 | | 18 | 4 | | 6 | 4 | | 5 | 4 | |
| | 2 | 38 | 48 | | 11 | 30 | | 17 | 10 | | 8 | 11 | | 4 | 4 | |
| | 3 | 102 | 91 | | 33 | 49 | | 53 | 18 | | 0 | 0 | | 8 | 13 | |
| **M category** | 0 | 205 | 194 | 0.731 | 84 | 137 | 0.573 | 89 | 25 | 0.204 | 11 | 11 | 1.00 | 21 | 21 | 0.509 |
| | 1 | 46 | 48 | | 11 | 23 | | 27 | 13 | | 4 | 5 | | 4 | 7 | |
| **UICC-Stage (8th eds.)** | I A/B | 42 | 41 | 0.787 | 25 | 37 | 0.435 | 15 | 1 | 0.108 | 0 | 0 | 0.877 | 2 | 3 | 0.406 |
| | II A/B | 56 | 51 | | 23 | 36 | | 23 | 8 | | 1 | 1 | | 9 | 6 | |
| | III A/B/C | 106 | 99 | | 36 | 62 | | 50 | 16 | | 10 | 9 | | 10 | 12 | |
| | IV | 46 | 48 | | 11 | 23 | | 27 | 13 | | 4 | 5 | | 4 | 7 | |
| **pL category** | 0 | 119 | 102 | 0.226 | 48 | 75 | 0.495 | 55 | 10 | 0.020 | 2 | 5 | 0.390 | 14 | 12 | 0.579 |
| | 1 | 114 | 123 | | 37 | 72 | | 53 | 26 | | 13 | 10 | | 11 | 15 | |
| **pV category** | 0 | 209 | 198 | 0.549 | 76 | 131 | 0.826 | 100 | 35 | 0.451 | 10 | 11 | 1.00 | 23 | 21 | 0.252 |
| | 1 | 232 | 225 | | 8 | 16 | | 8 | 1 | | 5 | 4 | | 2 | 6 | |
| **Tumor grade** | G1/G2 | 40 | 78 | <.001* | 39 | 74 | 0.363 | 1 | 2 | 0.153 | 15 | 16 | | 0 | 2 | 0.492 |
| | G3/G4 | 210 | 161 | | 56 | 83 | | 114 | 36 | | 15 | 16 | | 25 | 26 | |
| **R status** | 0 | 210 | 204 | 0.536 | 83 | 139 | 1.00 | 93 | 30 | 1.00 | 11 | 11 | 1.00 | 23 | 24 | 0.672 |
| | 1 | 33 | 27 | | 7 | 11 | | 20 | 7 | | 4 | 5 | | 2 | 4 | |
| **Microsatellite status** | MSS | 224 | 217 | 0.927 | 85 | 141 | 0.819 | 110 | 37 | nc | 12 | 16 | 0.448 | 17 | 23 | 0.339 |
| | MSI | 18 | 18 | | 9 | 13 | | 0 | 0 | | 1 | 0 | | 8 | 5 | |
| **EBV status** | neg. | 229 | 226 | 0.828 | 87 | 149 | 0.159 | 108 | 37 | 0.260 | 12 | 15 | 0.464 | 22 | 25 | 1.00 |
| | pos. | 12 | 10 | | 8 | 6 | | 0 | 1 | | 1 | 0 | | 3 | 3 | |
| **H. pylori** | neg. | 181 | 175 | 0.634 | 65 | 117 | 0.685 | 85 | 24 | 0.357 | 10 | 14 | 0.378 | 21 | 20 | 0.254 |
| | pos. | 29 | 32 | | 12 | 18 | | 11 | 6 | | 4 | 2 | | 2 | 6 | |
| **MET-status** | neg. | 235 | 219 | 0.155 | 90 | 151 | 1.00 | 109 | 32 | 0.029 | 12 | 13 | 1.00 | 24 | 23 | 0.113 |
| | pos. | 13 | 21 | | 5 | 8 | | 5 | 6 | | 3 | 3 | | 0 | 4 | |
| **HER2-status** | neg. | 212 | 198 | 0.165 | 71 | 126 | 0.680 | 103 | 29 | 0.012 | 14 | 15 | 1.00 | 24 | 28 | 0.472 |
| | pos. | 14 | 22 | | 12 | 15 | | 1 | 4 | | 0 | 1 | | 1 | 0 | |
| **PD-L1 in tumor cells** | neg. | 182 | 163 | 0.155 | 56 | 108 | 0.372 | 101 | 33 | 0.692 | 14 | 13 | 0.600 | 11 | 9 | 0.777 |
| | pos. | 49 | 61 | | 28 | 41 | | 6 | 3 | | 1 | 3 | | 14 | 14 | |

*(Continued)*

**Table 1.** (Continued)

| Cohort | | All | | | Intestinal | | | Diffuse | | | Mixed | | | Unclassified | | |
|---|---|---|---|---|---|---|---|---|---|---|---|---|---|---|---|---|
| | | | | | | | Tumor type according to Lauren | | | | | | | | | |
| FGFR status | | Neg. | Pos. | p | Neg. | Pos. | p | Neg. | Pos. | p | Neg. | Pos. | p | Neg. | Pos. | p |
| PD-L1 in TIL | ≤1 | 144 | 150 | 0.327 | 52 | 93 | 1.00 | 70 | 31 | 0.020 | 10 | 13 | 0.430 | 12 | 13 | 0.578 |
| | ≥1 | 87 | 74 | | 32 | 56 | | 37 | 5 | | 5 | 3 | | 13 | 10 | |
| VISTA status | neg. | 206 | 208 | 0.744 | 65 | 138 | 0.007 | 105 | 34 | .572 | 14 | 14 | 0.485 | 22 | 22 | 1.000 |
| | pos. | 22 | 19 | | 17 | 12 | | 2 | 2 | | 0 | 2 | | 3 | 3 | |

*Further significance after correcting the p-value using the exploratory Simes procedure for multiple testing. Prox., proximal; Dist., distal; MSS, microsatellite stable; MSI, microsatellite unstable; EBV, Epstein-Barr virus; TIL, tumor-infiltrating lymphocytes; nc, not calculable

The majority [240 cases (99.1%)] of FGFR2-positive GCs showed a variable combination of staining intensities (Fig 2). More than half [137 (56.6%)] of the cases showed even more than two different staining intensities, i.e., FGFR2-0/1$^+$ [83 cases (16.8%)], 0/2$^+$ [8 (1.6%)], 0/1$^+$/2$^+$ [40 (8.1%)], 0/3$^+$ [12 (2.4%)], 0/1$^+$/3$^+$ [2 (0.4%)], 0/2$^+$/3$^+$ [22 (4.5%)] or 0/1$^+$/2$^+$/3$^+$ [73 (14.8%)]. The percentage of the immunostained tumor area varied for all three staining intensities, e.g., ranging from 0 to 100% in the FGFR2-2$^+$ category. Furthermore, FGFR2 immunostaining was found at the invasion front of the tumor in 2.5% (6 cases), toward the gastric lumen in 19.4% (47 cases), and at the tumor center in 79.1% (189 cases) of the cases. Collectively, these data show that the expression (combination of intensity of immunostaining and amount of immunopositive tumor areas) of FGFR2 is heterogeneous in GC.

## Prognostic significance of FGFR2

Since we did not know a priori, which "cutoff" value of FGFR2 expression might be biologically relevant, we applied a stepwise explorative approach using OS and TSS as surrogates for a putative tumor biological significance.

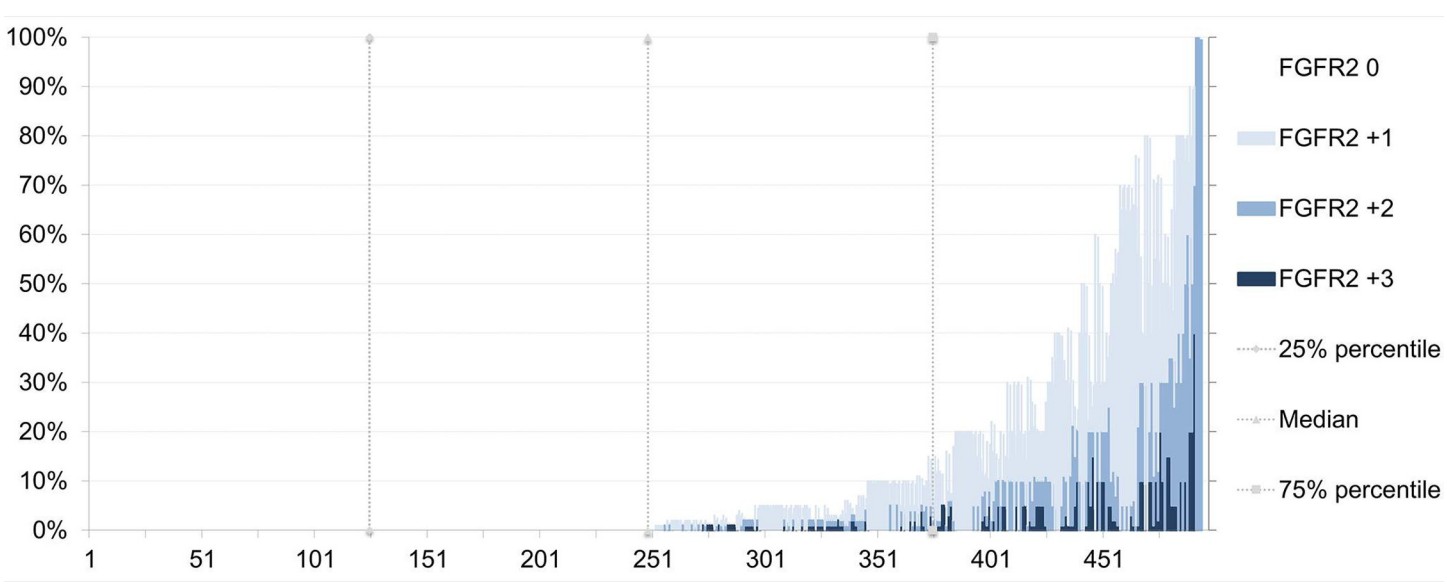

**Fig 2. Distribution of FGFR2 protein expression in the cohort.** FGFR2 staining was detected in 245 (49.1%) of the cases examined. Overall, there was a heterogeneous combination of different intensities of FGFR2 immunostaining. The occurrence of all staining intensities, i.e., FGFR-IHC 0/1$^+$/2$^+$ and 3$^+$ was observed in 73 cases (14.8%). Staining of large tumor fractions was observed rather rarely. Only 39 cases (7.9%) showed staining of more than 50% of the tumor cells.

Four different cutoff values were explored: HScore = 0 vs. HScore >0 (split at the median), FGFR-0/1$^+$ vs. FGFR-2$^+$/3$^+$, FGFR-3$^+$ absent vs. FGFR3$^+$ present, and above and below 95-per-centile of the HScore. As summarized in Table 2 and shown in Fig 3, none of the cutoff values correlated with OS or TSS in the entire patient cohort.

Next, we correlated FGFR2 status separately for the GC subtypes according to Laurén. Interestingly, OS and TSS of diffuse-type GC correlated significantly with FGFR2-status irrespective of the cutoff value. High FGFR2 expression in diffuse-type GC is associated with worse patient outcomes. No correlation was found with patient survival for the other phenotypes, that is, intestinal, mixed, and unclassifiable (Table 3 and Fig 4).

## Correlation of FGFR2 status with clinicopathological patient characteristics

FGFR2 status correlated with various patient characteristics (Table 1). Using the median HScore as the cutoff (0 vs. >0), a correlation analysis was performed for all cases. FGFR2 status was correlated with the Laurén phenotype and tumor grade. FGFR2 positive GCs were significantly more prevalent in intestinal-type GCs than in diffuse-type GCs. FGFR2 positive GC cases were found more frequently in grade 3 and 4 tumors. There were no significant correlations with the pT, pN, or pM categories (Table 1).

The correlation of the FGFR2-status with different patient characteristics was then assessed separately for each Lauren phenotype (Table 1). Using the median HScore as the cutoff, FGFR2 expression in intestinal-type GCs correlated inversely with VISTA status. The FGFR2 status of diffuse-type GC correlated with lymphatic invasion and MET and HER2 status, and inversely with PD-L1 expression in tumor-infiltrating immune cells. No significant correlation was found between FGFR2 status and any other patient characteristic.

Staining was also categorized as membranous or cytoplasmic (present vs. absent). We found no significant correlation between the intracellular localization of FGFR2 and any clinicopathological patient characteristic (S2 Table). In the diffuse-type GC we also tested whether

**Table 2. Analysis of patient survival using different definitions of FGFR2 positivity.**

| Cu-toff | | FGFR2-Positive | | FGFR2-Negative | | p-value | Cu-toff | | FGFR2-Positive | | FGFR2-Negative | | p-value |
|---|---|---|---|---|---|---|---|---|---|---|---|---|---|
| | | Events | (Cens.) | Events | (Cens.) | | | | Events | (Cens.) | Events | (Cens.) | |
| | | Median | (SD) | Median | (SD) | | | | Median | (SD) | Median | (SD) | |
| | | 95% Conf. Int. | | 95% Conf. Int. | | | | | 95% Conf. Int. | | 95% Conf. Int. | | |
| HScore >0 | OS | 230 | (47) | 243 | (53) | 0.254 | FGFR2-3$^+$ present | OS | 104 | (17) | 369 | (83) | 0.598 |
| | | 14.03 | (1.3) | 15.47 | (1.7) | | | | 14.65 | (1.5) | 14.98 | (1.4) | |
| | | 11.50/16.56 | | 12.23/18.71 | | | | | 11.64/17.66 | | 12.29/17.68 | | |
| | TSS | 230 | (70) | 243 | (67) | 0.813 | | TSS | 94 | (29) | 349 | (108) | 0.992 |
| | | 16,36 | (2.1) | 16.59 | (1.7) | | | | 15.64 | (2.2) | 16.59 | (1.7) | |
| | | 12.34/20.39 | | 13.29/19.90 | | | | | 11.38/19.90 | | 13.27/19.91 | | |
| FGFR2-2$^+$ or 3$^+$ present | OS | 152 | (28) | 321 | (72) | 0.245 | 0.95 percentile of the HScore | OS | 25 | (5) | 448 | (95) | 0.835 |
| | | 13.57 | (1.6) | 15.90 | (1.4) | | | | 10.51 | (11.5) | 14.88 | (1.1) | |
| | | 10.52/16.62 | | 13.17/18.63 | | | | | 0.00/32.96 | | 12.80/16.97 | | |
| | TSS | 138 | (42) | 305 | (95) | 0.475 | | TSS | 25 | (6) | 418 | (131) | 0.544 |
| | | 15.47 | (2.3) | 17.05 | (1.8) | | | | 20.04 | (11.7) | 16.53 | (1.4) | |
| | | 11.07/19.88 | | 13.46/20.64 | | | | | 0.00/42.89 | | 13.89/19.17 | | |

OS: Overall survival, TSS: Tumor specific survival, time in months

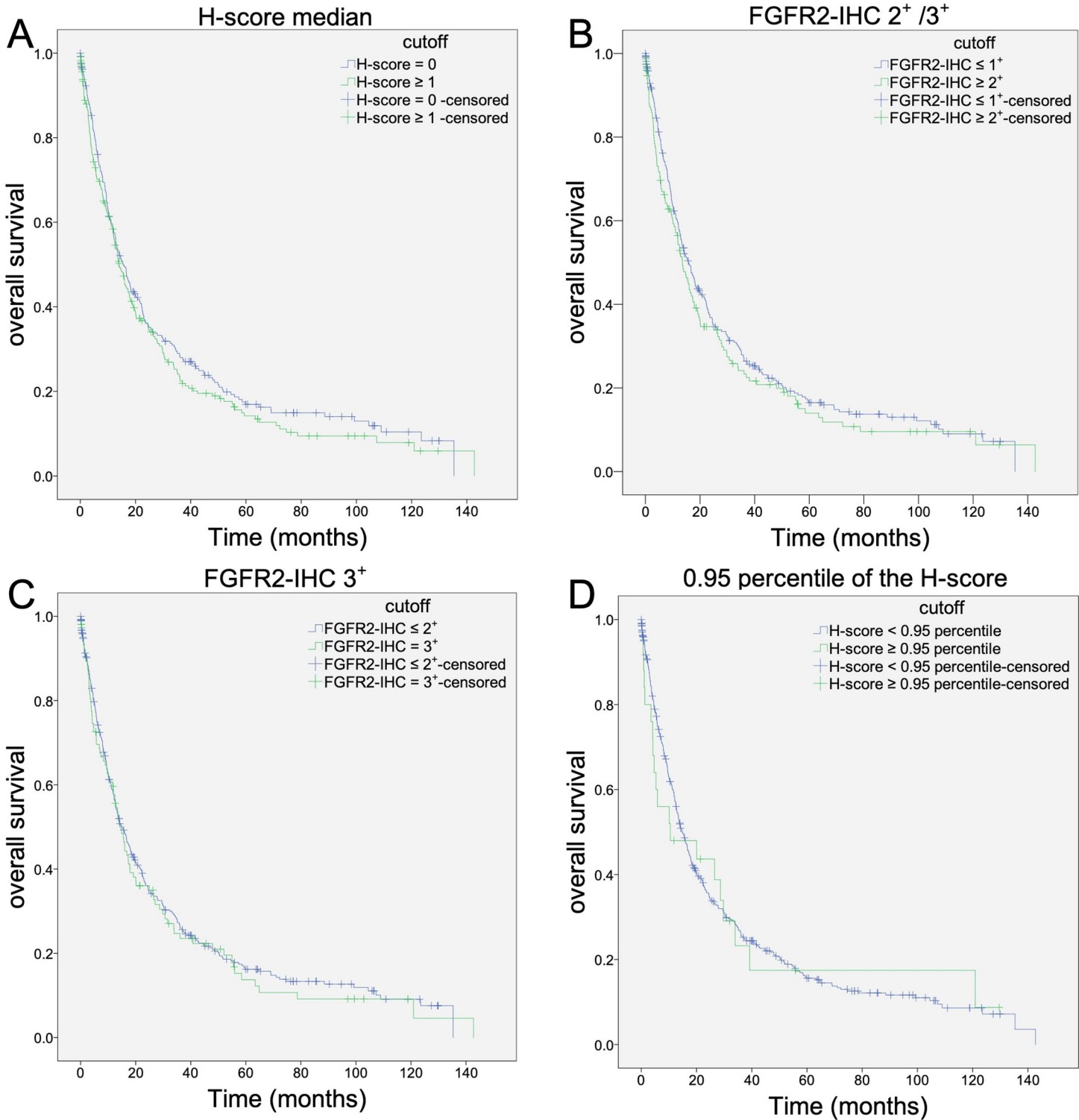

**Fig 3. Analysis of patient overall survival using different definitions of FGFR2 positivity in the immunohistochemical examinations.** Analysis of survival data was first performed for the entire cohort. Using different thresholds to define increased FGFR2 protein expression, there was no difference in overall or tumor-specific patient survival. Initially, the median Hscore was used as a cutoff (A). The presence of FGFR2-IHC 2+ or 3+ (B) and the exclusive presence of FGFR2-IHC 3+ (C) were also examined. Finally, the 0.95 percentile of the Hscore was used as a cutoff (D).

**Table 3. Analysis of patient survival using different definitions of FGFR2 positivity separately according to Laurén classification.**

| Tumor type | | | Intestinal | | | | Diffuse | | | | Mixed | | | | Unclassified | | |
|---|---|---|---|---|---|---|---|---|---|---|---|---|---|---|---|---|---|
| FGFR2-status | | | Positive | Negative | p | | Positive | Negative | p | | Positive | Negative | p | | Positive | Negative | p |
| Cutoff | | | Events (Cens.) / Median (SD) / 95% Conf. Int. | Events (Cens.) / Median (SD) / 95% Conf. Int. | | | Events (Cens.) / Median (SD) / 95% Conf. Int. | Events (Cens.) / Median (SD) / 95% Conf. Int. | | | Events (Cens.) / Median (SD) / 95% Conf. Int. | Events (Cens.) / Median (SD) / 95% Conf. Int. | | | Events (Cens.) / Median (SD) / 95% Conf. Int. | Events (Cens.) / Median (SD) / 95% Conf. Int. | |
| **HScore median** | OS | | 117 (36) / 15.64 (2.1) / 11.47/19.81 | 71 (20) / 17.91 (3.4) / 11.32/24.50 | 0.438 | | 32 (2) / 7.98 (6.3) / 0.00/20.19 | 91 (22) / 14.98 (3.4) / 10.60/19.36 | 0.015 | | 14 (2) / 6.74 (4.8) / 0.00/16.14 | 12 (3) / 9.99 (5.0) / 0.12/19.86 | 0.438 | | 20 (7) / 16.00 (4.9) / 6.36/25.64 | 16 (8) / 19.61 (7.5) / 4.97/34.26 | 0.342 |
| | TSS | | 90 (51) / 18.99 (3.9) / 11.44/26.54 | 61 (26) / 17.97 (6.5) / 5.28/30.66 | 0.864 | | 27 (5) / 5.68 (1.8) / 2.08/9.28 | 80 (28) / 15.47 (2.0) / 11.52/19.43 | 0.021 | | 10 (2) / 6.74 (4.2) / 0.00/14.88 | 12 (3) / 9.99 (5.0) / 0.12/19.86 | 0.591 | | 14 (12) / 16.79 (4.7) / 7.59/25.99 | 10 (10) / 24.41 (14.8) / 0.00/53.34 | 0.797 |
| **0.95 percentile of the HScore** | OS | | 99 (30) / 15.47 (2.2) / 11.12/19.83 | 89 (26) / 17.97 (5.0) / 8.2/27.74 | 0.344 | | 26 (1) / 7.98 (5.6) / 0.00/18.96 | 97 (23) / 14.62 (1.9) / 10.88/18.36 | 0.019 | | 12 (2) / 6.74 (4.9) / 0.00/16.37 | 14 (3) / 9.99 (5.4) / 0/20.59 | 0.430 | | 17 (6) / 16.00 (4.4) / 7.34/24.66 | 19 (9) / 19.61 (4.6) / 10.56/28.67 | 0.521 |
| | TSS | | 75 (42) / 18.76 (4.1) / 10.64/26.88 | 76 (35) / 18.27 (6.9) / 4.74/31.79 | 0.647 | | 21 (4) / 5.68 (2.0) / 1.69/9.68 | 86 (29) / 14.69 (1.9) / 10.93/18.45 | 0.033 | | 9 (2) / 9.30 (3.6) / 2.26/16.33 | 13 (3) / 4.53 (2.7) / 0.00/9.90 | 0.744 | | 11 (11) / 16.70 (12.2) / 0.00/40.71 | 15 (11) / 19.88 (7.1) / 5.90/33.86 | 0.814 |
| **IHC 2+ or 3+ present** | OS | | 82 (21) / 15.47 (2.4) / 10.72/22.23 | 106 (35) / 17.97 (3.5) / 11.04/24.90 | 0.401 | | 18 (1) / 5.29 (2.5) / 0.44/10.14 | 105 (23) / 14.62 (1.9) / 10.88/18.36 | 0.021 | | 9 (1) / 4.04 (3.0) / 0.00/10 | 17 (4) / 9.99 (4.6) / 0.89/19.09 | 0.374 | | 15 (5) / 16.00 (4.5) / 7.14/24.87 | 21 (10) / 19.61 (7.9) / 4.14/35.09 | 0.357 |
| | TSS | | 61 (32) / 17.94 (4.7) / 8.65/27.22 | 90 (45) / 18.43 (4.9) / 8.91/27.95 | 0.865 | | 17 (1) / 5.29 (1.3) / 2.81/7.77 | 90 (32) / 14.67 (1.9) / 10.93/18.45 | 0.004 | | 6 (1) / 4.04 (3.6) / 0.00/11.04 | 16 (4) / 9.99 (4.6) / 1.07/18.91 | 0.281 | | 12 (8) / 16.36 (4.2) / 8.12/24.60 | 14 (14) / 24.41 (16.2) / 0.00/56.15 | 0.368 |
| **Algorithm 1** | OS | | 107 (36) / 17.97 (3.57) / 10.98/24.95 | 61 (23) / 15.47 (2.26) / 11.04/19.91 | 0.296 | | 10 (0) / 2.99 (1.70) / .00/6.32 | 108 (23) / 14.69 (1.88) / 10.99/18.38 | 0.224 | | 6 (1) / 13.57 (8.95) / .00/31.11 | 17 (4) / 9.99 (4.64) / .89/19.09 | 0.911 | | 11 (3) / 11.89 (6.09) / .00/23.82 | 21 (10) / 19.61 (7.89) / 4.14/35.09 | 0.504 |
| | TSS | | 44 (23) / 17.94 (2.79) / 12.47/23.41 | 91 (46) / 18.43 (5.05) / 8.83/28.33 | 0.810 | | 9 (0) / 2.99 (0.39) / 2.22/3.76 | 93 (32) / 14.98 (1.78) / 11.49/18.47 | 0.071 | | 3 (1) / 6.74 (6.47) / .00/19.42 | 16 (4) / 9.99 (4.55) / 1.07/18.91 | 0.989 | | 9 (5) / 11.89 (6.30) / .00/24.24 | 14 (14) / 24.41 (16.20) / .00/56.15 | 0.457 |

OS: Overall survival, TSS: Tumor-specific survival, time in months, Algorithm 1: All cases with either FGFR-IHC 3+ or FGFR2-IHC 2+ and amplification in *Fgfr2*-CISH were declared positive.

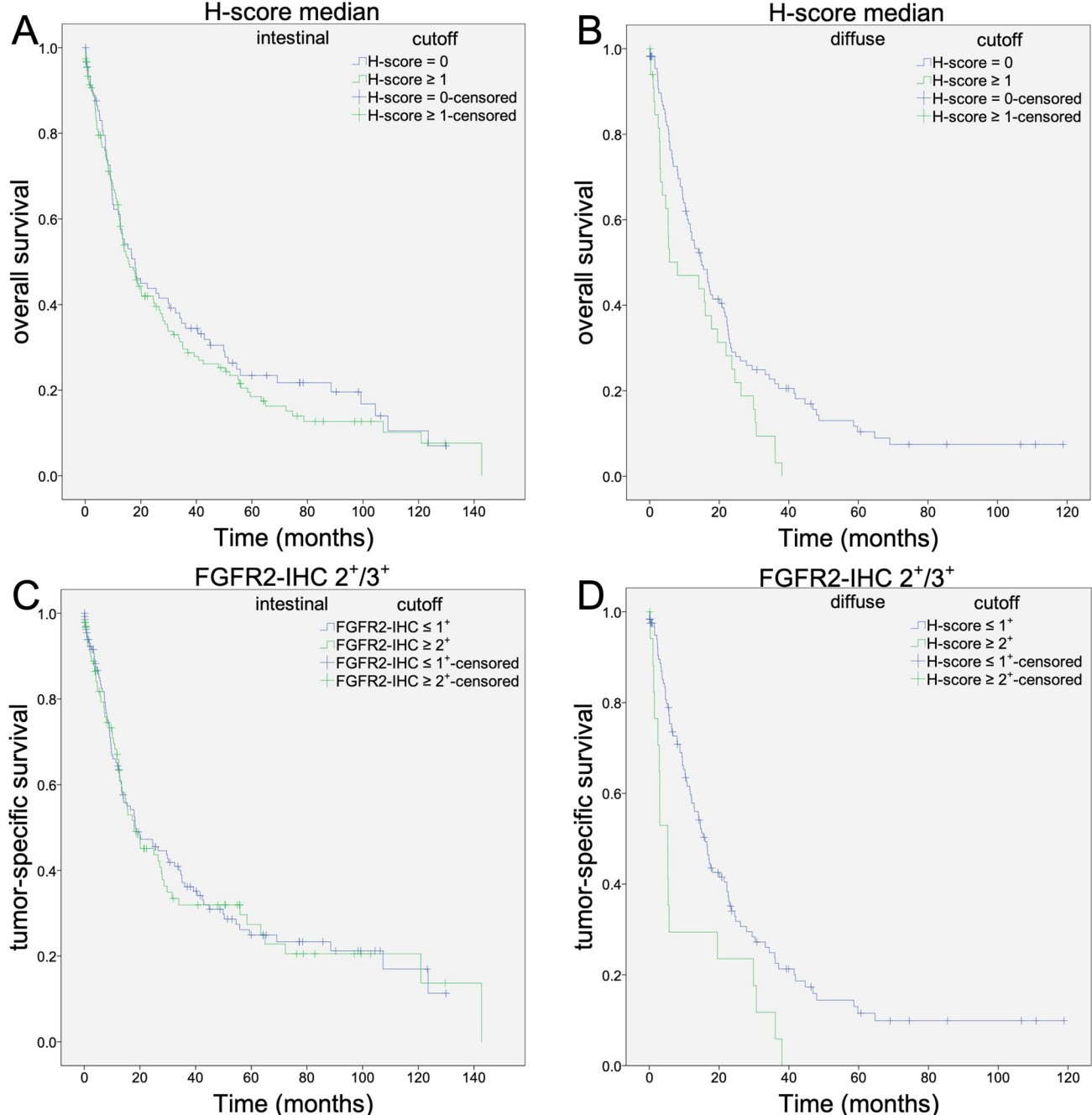

**Fig 4. Analysis of patient overall- and tumor-specific survival using different definitions of FGFR2 positivity according to Laurén classification.** Further analysis of survival time was performed separately according to the subgroups of the Laurén classification. Initially, the median Hscore was also used as a cutoff (A-B). For the intestinal type (A), there was no difference. For the diffuse-type (B), significantly worse overall and tumor specific survival was seen for patients with increased FGFR2 expression. The same trend was seen when using the presence of FGFR2-IHC 2+ or 3+ (C-D) as a cutoff. Here, also in patients with intestinal-type carcinoma (C), there was no difference in overall or tumor-specific patient survival. In patients with diffuse-type gastric carcinoma (D) and increased FGFR2 expression, significantly worse overall and tumor-specific survival was observed.

a higher pL category was associated with distant metastases (pM category), and no significant correlation was found (S3 Table).

## Chromogenic in situ hybridization provides evidence of intratumoral heterogeneity

Previously, we demonstrated that the expression of MET and HER2 shows substantial intratumoral heterogeneity, which is not only applicable to protein expression, as detected by immunohistochemistry, but also to gene amplification. Finally, we were interested in confirming that intratumoral heterogeneity also applies to *Fgfr2* amplification in GC patients.

Because of the heterogeneous distribution of the 108 cases with strong immunostaining (FGFR2-3[+]) and 50 cases of those with only 1% or less expression of FGFR2-3[+], we decided to evaluate all cases with ≥5% FGFR2-3[+] or ≥15% FGFR2-2[+] using CISH. In total, CISH was performed in 50 cases that met the immunohistochemical inclusion criteria. Seven cases could not be evaluated because of poor CISH or sample quality. In addition, 15 cases were examined using CISH due to their heterogeneous distribution in FGFR2-IHC. The average *Fgfr2* signal ranged from 1.75 to 32.9 copies per cell, with a mean copy number of 6.01. The average centromere 10 signal ranged from 1.55 to 4.55 copies per cell with a mean signal number of 2.34. Clustering of the *Fgfr2* signals was observed in 18 cases. Using the *Fgfr2*/centromere 10 copy number ratio of >2.2 as a cutoff, 20 cases were defined to be *Fgfr2* amplified. The *Fgfr2*/centromere 10 ratio ranged from 0.78 to 13.16 with a mean of 2.64. Twenty-seven cases with an average *Fgfr2* copy number of >4 were defined as *Fgfr2* polysomics. Gene amplification was heterogeneous, in that amplified and unamplified tumor areas were sharply demarcated on a cell-by-cell basis (Fig 1). Unspecific CISH-colored cells, as shown in Fig 1, were found in 14 cases. These data show that intratumoral heterogeneity also applies to *Fgfr2* gene amplification.

Based on comprehensive molecular analysis, the Cancer Genome Atlas Research Network proposed four molecular subgroups of GC: EBV+, MSI, chromosomal instability, and genomically stable GC [5]. Previously, amplification of *HER2* and *MET* was primarily observed in chromosomal unstable GC, which frequently harbors an intestinal phenotype [5]. We then correlated *Fgfr2*-amplification with phenotype according to Lauren and found that of the cases examined for *Fgfr2*-amplification, 35 showed intestinal-type GC and 12 showed diffuse-type GC. *Fgfr2*-amplification was detectable in 10 of the 35 patients with intestinal-type GC and in seven of the 12 patients with diffuse-type GC. Three patients with evidence of *Fgfr2*-amplification showed a positive MET status. A positive HER2 status was not observed in any of the patients with *Fgfr2*-amplification.

## Discussion

FGFR2 is involved in numerous physiological functions including cell proliferation, survival, migration, and angiogenesis. It is regularly expressed in many tissues and is susceptible to dysregulation in cancer cells. The effect of FGFR2 pathway activation is context-dependent and can evoke oncogenic and tumor-suppressive effects. Several mechanisms of genetic alterations, such as gene amplification, activating mutations, chromosomal translocations, single nucleotide polymorphisms, and aberrant splicing at the post-transcriptional level have been described. The majority of genomic aberrations lead to constitutive receptor activation and ligand-independent signaling. Auto- and paracrine activation might be important as well [37]. Amplifications and mutations of *Fgfr2* have been reported in breast, endometrial, and GC tissues [38].

In our study, we aimed to validate findings from previous studies focusing on White patients, since the majority of currently available data on FGFR2 in GC were derived from Asian study populations. A meta-analysis published in 2019 on the prognostic significance of FGFR2 protein expression, which included 4294 patients with GC, primarily summarized data

from the Japanese, South Korean, and Chinese patient cohorts (S4 Table) [28]. Of the ten studies reviewed, only two included partial data from non-Asian populations [16, 25]. The exact number of White patients could not be determined from the information provided by the authors [28]. Data from a white cohort regarding FGFR2 protein expression in GC could not be found by the authors in their literature search [28]. This difference in research focus is understandable given the much higher incidence and prevalence of GC in Asia [2, 39]. However, this disparity necessitates independent validation studies on white patient populations, as GC in Asians and Whites differs in terms of phenotype and prognosis: the intestinal phenotype is more common in White patients [11] and Asian GC patients have a much more favorable prognosis. To the best of our knowledge, this is the first study to examine FGFR2 expression in GC in a large central European cohort.

The interpretation of immunostaining results is challenging, and the use of different primary antibodies, staining protocols, and evaluation schemes may compromise comparability and data interpretation [21]. Therefore, we used different cutoff values for the assessment of the FGFR2 status (Table 2): overexpression ranged from 21.9% to 32.1%, which is consistent with previous findings ranging from 4% to 60% [17–27]. Using the same cutoff value (presence of an FGFR2$^+$-3$^+$), the prevalence of increased FGFR2 expression in the studied cohort was 22%, and slightly lower than the 31.9% previously reported by Hosoda et al. [27]. This indicates that the overall prevalence of FGFR2 expression was similar in the Asian and white cohorts.

Data on the association between FGFR2 expression and Lauren classification did not show a consistent picture so far [13, 39, 40]. Overexpression of FGFR2 was reported to be associated with poorly differentiated GC, which falls into the category of diffuse-type GC [40]. Again, these data were obtained from the Chinese cohort. In contrast, in our cohort, we found a significant correlation between FGFR2 expression and the intestinal tumor type (p<0.001).

Previous investigations have reported that increased FGFR2 expression is associated with a worse prognosis of long-term survival in all patients with GC [12, 20–23, 28]. However, the literature on FGFR2 also includes studies in which FGFR2 expression status was not associated with worse prognosis for all patients, but a shortened survival time was demonstrated exclusively in a subgroup of the Laurén classification [26, 27]. Likewise, using different cutoff values, we were unable to find a correlation between FGFR2 expression and patient outcomes in the entire cohort (Table 2 and Fig 4). However, we show that the biological significance of FGFR2 is a function of tumor type according to the Lauren type. High FGFR2 levels are only prognostically relevant in diffuse-type GCs. These observations are in line with the findings of Inokuchi et al., who also demonstrated the prognostic significance of FGFR2 expression in patients with diffuse-type GC [26]. Assessment of the tumor biological function of FGFR2 necessitates consideration of the tumor phenotype.

Regarding the association with other clinicopathological patient characteristics, FGFR2 status was reported to be correlated with the depth of tumor invasion, higher rate of lymph node metastasis, and more advanced stage [28]. In line with these findings, FGFR2-status was linked to lymph vessel invasion in diffuse-type GC and may contribute to disease progression. However, no correlation between lymph vessel invasion and the presence of distant metastasis was found in patients with diffuse GC (S3 Table). These observations highlight the difficulty of interpreting the biological significance of FGFR2 in GC.

Nevertheless, a meta-analysis by Kim et al. showed a significant correlation between high FGFR2 expression and depth of tumor invasion, higher rate of lymph node metastasis, more advanced disease stage, and significantly worse survival [28]. These observations were made for all patients with GC without separating them into further subgroups. In our study, we only demonstrated a correlation between increased lymphatic invasion and worse survival in patients with FGFR2-positive diffuse-type GC using different cutoff values. We did not detect

any effects in any of the patients. One explanation for this discrepancy might be the different prevalence of GC phenotypes in different patient cohorts. In support of this contention, the Asian study cohorts included a higher proportion of diffuse-type GC compared with our cohort, i.e., 53.7% vs. 31.4% (S4 Table) [28].

Sampling error was another confounding factor. As shown previously for HER2, the use of tissue microarrays (TMA) instead of large-area tissue sections carries the risk of sampling errors when assessing TRK expression in GC [7]. Given the significant intratumoral heterogeneity, as shown again here for FGFR2, the use of TMAs may lead to both over- and underestimation and hence non-representative prevalence [7]. We reduced the risk of sampling errors using large-area tissue sections.

The heterogeneity of malignant tumors is a major barrier to drug development and long-term disease control. They can be categorized into intertype heterogeneity (differences between the cancers of two patients, each with a different tumor type), intratype heterogeneity (cancers of the same type differ in two different individuals), intraprimary heterogeneity (genetic heterogeneity between two cells of the same primary tumor), intermetastatic heterogeneity (genetic heterogeneity between cells of different metastases), and intrametastatic heterogeneity (genetic heterogeneity between two cells of the same metastasis). The mechanisms of tumor heterogeneity are diverse and complex and also apply to GC [41]. They enclose tumor evolution and adaptation to diverse environmental constraints, including chemotherapy [42–44]. Regarding FGFR2, we found evidence of intratype and intraprimary heterogeneity on the genetic (chromogenic in situ hybridization) and expression level (immunohistochemistry), which may compromise accurate assessment of FGFR2 status, as has been shown for HER2. Thus, testing for FGFR2, for example, as a predictive biomarker, requires consideration of testing algorithms that have been developed for HER2-testing [45].

Despite recent advances, the results of chemotherapy for GC treatment remain unsatisfactory [46]. Owing to its biological importance, the FGFR2 receptor represents a potential target for the development of new therapies for GC [37]. Clinical studies on the effect of FGFR2 inhibitors show a mixed picture of the effectiveness of treatment in GC patients [47, 48]. Some authors attributed these observations to the misselection of patients [49]. For example, analysis of data on the FGFR inhibitor AZD4547 showed that only some subgroups of *Fgfr2* amplified tumors were responsive to therapy [50]. This observation highlights the need for better characterization of the subpopulations of tumors such as GC to identify patients more accurately for targeted drug therapy in the future. The observation that increased FGFR2 expression in our patient cohort was exclusively of prognostic significance in patients with diffuse-type GC should be considered in the future when evaluating trial data on FGFR2 inhibitors.

Previous studies on the association of FGFR2 with other TKRs in GC showed that common gene amplification of TKRs, such as *Fgfr2*, *HER2*, *MET*, and *EGFR*, are mutually exclusive [51, 52]. Notably, in isolated cases, it has been reported that amplification of *Fgfr2* occurs in one part of the tumor and amplification of *HER2* is detectable in another part of the tumor. However, amplification of different TKR genes in the same tumor cells has not been described to date [11]. Furthermore, the protein expression showed a different pattern. Thus, increased protein expression of FGFR2 does not exclude the expression of other TKRs [22]. Likewise, it has been reported that despite gene amplification of one TKR, increased protein expression of other TKRs on the cell surface is possible [14, 53]. Patients with *Fgfr2* amplification and concomitant overexpression of MET have been described [14]. In our cohort, a correlation between FGFR2 and MET expression was observed (p = 0.029, Table 3), although these results were not significant after adjusting the p-value for multiple testing. However, these observations indicate that, unlike amplifications, increased protein expression of FGFR2 and other TKRs, such as MET, may occur in GC.

Our study has some limitations. First, the data were obtained only from a single center, which may limit the transferability to other white patient populations. A multicenter study is necessary to validate our data. Second, we did not study the entire cohort for *Fgfr2* amplifications and could not comment on the combined association of FGFR2 protein expression and *Fgfr2*-amplification with clinicopathological patient characteristics. However, genomic changes of *Fgfr2* known to be oncogenic include amplifications, short variants, and rearrangements, constituting 72%, 13%, and 8.6% of the *Fgfr2*-alterations in GC [54]. Thus, correlating genetic alterations of *Fgfr2* with clinicopathological patient characteristics necessitates a more comprehensive analysis and is beyond the scope of the current study.

## Conclusion

In summary, our study on a large and well-characterized White patient population showed that FGFR2 is expressed heterogeneously in GC, partly related to heterogeneous *Fgfr2* amplification, sharing features with other TKRs, such as HER2 or MET. The biological significance of FGFR2 is a function of tumor type according to Lauren and predicts poor patient outcome in diffuse-type GC in White patients. Differences with data obtained in Asian patient populations are related to different prevalence of tumor types, that is, intestinal vs. diffuse-type GC, and the overall better outcome of GC in Asian patient populations.

## Supporting information

**S1 Table. Primary data of FGFR2 immunohistochemistry, CISH and clinicopathological patient characteristics.**
(XLSX)

**S2 Table. Association of clinicopathological patient characteristics of diffuse-type gastric cancer differentiated by cellular localization of FGFR2-immunostaining.**
(XLSX)

**S3 Table. Association of pL-category and pM-category in patients with diffuse-type gastric cancer.**
(XLSX)

**S4 Table. Occurrence of diffuse and intestinal tumor types regarding FGFR2 expression und studies investigated by Kim et al. [28].**
(XLSX)

## Author Contributions

**Conceptualization:** Jochen Haag, Sandra Krüger, Christoph Röcken.

**Data curation:** Thorben Schrumpf, Hans-Michael Behrens, Sandra Krüger.

**Formal analysis:** Thorben Schrumpf, Hans-Michael Behrens, Jochen Haag.

**Investigation:** Thorben Schrumpf, Jochen Haag.

**Methodology:** Thorben Schrumpf, Hans-Michael Behrens, Jochen Haag, Sandra Krüger, Christoph Röcken.

**Project administration:** Sandra Krüger, Christoph Röcken.

**Resources:** Christoph Röcken.

**Software:** Hans-Michael Behrens, Christoph Röcken.

**Supervision:** Christoph Röcken.

**Validation:** Thorben Schrumpf, Jochen Haag, Sandra Krüger.

**Visualization:** Thorben Schrumpf, Christoph Röcken.

**Writing – original draft:** Thorben Schrumpf, Hans-Michael Behrens, Jochen Haag, Sandra Krüger, Christoph Röcken.

**Writing – review & editing:** Thorben Schrumpf, Hans-Michael Behrens, Jochen Haag, Sandra Krüger.

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
