## [Decision Letter · Decision Letter 0]

2 Nov 2021

PONE-D-21-21424FGFR2 overexpression is associated with shorter survival in diffuse type gastric cancer in a large Central European cohort.PLOS ONE

Dear Dr. Röcken,

Thank you for submitting your manuscript to PLOS ONE. After careful consideration, we feel that it has merit but does not fully meet PLOS ONE’s publication criteria as it currently stands. Therefore, we invite you to submit a revised version of the manuscript that addresses the points raised during the review process. Please pay a careful attention to the crical comments of the Reviewer. Specifically, discuss the results obtained more thoroughly in comparison with other, published studies including a large meta-analysis referred to as was the promary aim of the resesearch. Also, please consider and discuss the potential clinical implications of your main finding of the negative correlation of FGFR2 expression to patient outcome in diffuse gastric cancer. Also, could any correlation of FGFR2 status to clinical progression of gastric cancer patients be found? Did observed lymphatic invasion correlate with the metastatic pattern of diffuse gastric cancer? Mechanistically, the observed heterogeneity of FGFR2 staining could be discussed in more detailed way. Did cellular compartmentalisation (membrane staining vs intracellular staining analysed in the study) of the FGFR2 IHC correlate with any characteristics of diffuse gastric cancer compared to intestinal type.  Additionally, the cohort from the Kiel area may not represent the whole European population although certainly to it. The term "caucasian" may not be appropriate.  In the end, I apologize for the delayed answer, caused by cancelled reviewer commitments and other unfortunate delays in the processing of the manuscript.

If you consider resubmission, please submit the revised manuscript by November 31, 2021. If you will need more time than this to complete your revisions, please reply to this message or contact the journal office at plosone@plos.org. Please include the following items when submitting your revised manuscript:A rebuttal letter that responds to each point raised by the academic editor and reviewer(s). You should upload this letter as a separate file labeled 'Response to Reviewers'.A marked-up copy of your manuscript that highlights changes made to the original version. You should upload this as a separate file labeled 'Revised Manuscript with Track Changes'.An unmarked version of your revised paper without tracked changes. You should upload this as a separate file labeled 'Manuscript'.

We look forward to receiving your revised manuscript.

Kind regards,

Pirkko L. Härkönen, M.D., Ph.D.

Academic Editor

PLOS ONE

Journal Requirements:

2. Please note that according to our submission guidelines (http://journals.plos.org/plosone/s/submission-guidelines), outmoded terms and potentially stigmatizing labels should be changed to more current, acceptable terminology. For example: “Caucasian” should be changed to “white” or “of [Western] European descent” (as appropriate)

Reviewers' comments:

Reviewer's Responses to Questions

**Comments to the Author**

1. Is the manuscript technically sound, and do the data support the conclusions?

Reviewer #1: Partly

2. Has the statistical analysis been performed appropriately and rigorously? 

Reviewer #1: Yes

3. Have the authors made all data underlying the findings in their manuscript fully available?

Reviewer #1: No

4. Is the manuscript presented in an intelligible fashion and written in standard English?

Reviewer #1: Yes

5. Review Comments to the Author

Reviewer #1: The study analyzes FGFR2 expression in gastric cancer using immunohistochemistry and chromogenic-in-situ hybridization techniques. It comes from the Department of Pathology at Christian-Albrechts- University in Kiel, Germany. The list of authors does not seem to include any surgeons, oncologists nor epidemiologists.

The FGFR2 expression was found to be very heterogenous. The FGFR2 protein expression did not correlate with patient survival in the entire cohort. After subanalysis, the authors found a correlation between FGFR2 expression and patient outcome in diffuse type gastric cancer. The subset of patients (FGFR2-positive and diffuse type) constitutes a small subset of patients, as the authors conclude. Further, the main finding of the study with Central European patient cohort really was that FGFR2 overexpression is associated with shorter survival in diffuse type of gastric cancer. The authors do not elaborate this further. So what? What are the clinical implication? Differences in Asia vs. Europe?

In the literature, there several reports on the correlation (or lack of it) between FGFR2 expression and patient survival. This study, unfortunately, does not clear the discrepancy between previous findings.

In several parts of the manuscript, linguistic revision is needed.

Minor comments:

Line 81: The coverage and reliability of collecting follow-up data from interviews of general practitioners remain limited.

Figure 1 is lacking the bars.

6. PLOS authors have the option to publish the peer review history of their article (what does this mean?). If published, this will include your full peer review and any attached files.

Reviewer #1: No

---

## [Author Response · Author response to Decision Letter 0]

26 Jan 2022

Dear Sir or Madam,

please find attached our revised manuscript, entitled “FGFR2 overexpression and compromised survival in diffuse-type gastric cancer in a large central European cohort”, which we are submitting to PlosOne.

We thank the editorial board for giving us the chance to revise our manuscript and also want to thank the reviewer for the critical comments. We have implemented changes to our manuscript based on the comments and believe this has improved the quality of our manuscript. All changes are highlighted in red font.

Editorial comment:

Please pay a careful attention to the critical comments of the Reviewer. 

Response: All reviewer comments were considered. We also added four Supplemental Ta-bles, including a Supplemental Table with the minimal data set (Table S1).

Specifically, discuss the results obtained more thoroughly in comparison with other, published studies including a large meta-analysis referred to as was the primary aim of the research. 

Also, please consider and discuss the potential clinical implications of your main finding of the negative correlation of FGFR2 expression to patient outcome in diffuse gastric cancer. 

Response: The discussion was amended, including an additional section on the clinical signifi-cance of our findings. 

Also, could any correlation of FGFR2 status to clinical progression of gastric cancer patients be found? Did observed lymphatic invasion correlate with the metastatic pattern of diffuse gas-tric cancer? 

Response: As shown in Table 1, T category and UICC tumor stage as surrogates for tumor progression, did not correlate with the FGFR status. We also carried out additional statistics. However, no correlation was found between pL- and pM category in diffuse type GC (Table S3). 

Mechanistically, the observed heterogeneity of FGFR2 staining could be discussed in more detailed way.

Response: We have added a section on tumor heterogeneity in the discusson. We were able to demonstrate in our work signs of intratype and intraprimary heterogeneity at the genetic lev-el as well as that in protein expression of FGFR2.

Did cellular compartmentalisation (membrane staining vs intracellular staining analysed in the study) of the FGFR2 IHC correlate with any characteristics of diffuse gastric cancer compared to intestinal type. 

Response: We performed additional statics and did not find any corelation between intracellular localization of FGFR2 immunostaining any clinicopathological patient characteristic (Table S3). The data were added to the revised manuscript.

Additionally, the cohort from the Kiel area may not represent the whole European population although certainly to it. 

Response: We added the following sentence to the discussion: “Our study has some limita-tions. First, the data were obtained only from a single center, which may limit the transferability to other white patient populations. A multicenter study is necessary to validate our data.”

The term "caucasian" may not be appropriate. 

Response: We acknowledge the recent discussion on the terms “Caucasian” and “White". The terms “European” and “Caucasian” were replaced by “White” where appropriate.

Reviewer #1: 

1) The study analyzes FGFR2 expression in gastric cancer using immunohistochemistry and chromogenic-in-situ hybridization techniques. It comes from the Department of Pa-thology at Christian-Albrechts-University in Kiel, Germany. The list of authors does not seem to include any surgeons, oncologists nor epidemiologists.

Response: The authors of our manuscript adhere to the criteria based on the International Committee of Medical Journal Editors, i.e., (1) substantial contributions to conception and de-sign, acquisition of data, or analysis and interpretation of data; (2) drafting the article or revising it critically for important intellectual content; (3) final approval of the version to be published, and

(4) agreement to be accountable for all aspects of the work in ensuring that questions related to the accuracy or integrity of any part of the work are appropriately investigated and resolved. No surgeon, oncologist or epidemiologist fulfilled these criteria to justify co-authorship.

2) The FGFR2 expression was found to be very heterogenous. The FGFR2 protein ex-pression did not correlate with patient survival in the entire cohort. After subanalysis, the authors found a correlation between FGFR2 expression and patient outcome in diffuse type gastric cancer. The subset of patients (FGFR2-positive and diffuse type) consti-tutes a small subset of patients, as the authors conclude. Further, the main finding of the study with Central European patient cohort really was that FGFR2 overexpression is associated with shorter survival in diffuse type of gastric cancer. The authors do not elaborate this further. So what? What are the clinical implication? Differences in Asia vs. Europe?

Response: We amended the discussion.

3) In the literature, there several reports on the correlation (or lack of it) between FGFR2 expression and patient survival. This study, unfortunately, does not clear the discrepan-cy between previous findings.

Response: As mentioned above in response to the editorial comments, the discussion was re-vised and an additional section on the clinical implications of our work was added. We have also attempted to provide a clearer comparison of the differences between our European and the previously studied Asian cohorts. The leading differences we see are the different distribution of tumor types and the better prognosis of the Asian patients.

4) In several parts of the manuscript, linguistic revision is needed.

Response: The manuscript was proof read by Whiley editing service. However, professional editing focused solely on linguistic errors and grammar and not on the scientific content.

Minor comments:

Line 81: The coverage and reliability of collecting follow-up data from interviews of general practitioners remain limited.

Response: We agree with the referee. 

Figure 1 is lacking the bars.

Response: The bars were added to the figure.

Additional comments: 

- During revision we noticed that we had used the 7th edition of UICC, which is outdated. We therefore revised statistics using the 8th edition of the TNM Classification of Malig-nant Tumors. The revision did not result in any significant changes.

- We also reworded the title to make it more concise.

We confirm that the publication is approved by all authors. The authors hereby state that the manuscript has not been published and that, if accepted, it will not be published elsewhere in the same form, in English or in any other language, including electronically without the written consent of the copyright-holder. All authors have contributed to the work described sufficiently to be named as authors. We hereby confirm that all presented material is original research. The authors declare no conflict of interest.

In looking forward to your response

Yours sincerely

Prof. Dr. med. C. Röcken

---

## [Editor Report · Decision Letter 1]

2 Feb 2022

FGFR2 overexpression and compromised survival in diffuse-type gastric cancer in a large central European cohort.

PONE-D-21-21424R1

Dear Dr. Röcken,

We’re pleased to inform you that your manuscript has been judged scientifically suitable for publication and will be formally accepted for publication once it meets all outstanding technical requirements.

The revised manuscript addresses the concerns of the previous Reviews in an appropriate way. The manuscript has greatly improved. 

Kind regards,

Pirkko L. Härkönen, M.D., Ph.D.

Academic Editor

PLOS ONE
---

## [Editor Report · Acceptance letter]

4 Feb 2022

PONE-D-21-21424R1 

FGFR2 overexpression and compromised survival in diffuse-type gastric cancer in a large central European cohort 

Dear Dr. Röcken:

I'm pleased to inform you that your manuscript has been deemed suitable for publication in PLOS ONE. Congratulations! Your manuscript is now with our production department. 

Kind regards, 

on behalf of

Dr. Pirkko L. Härkönen 

Academic Editor

PLOS ONE